# In Vitro Activity of Delafloxacin Against *Corynebacterium* spp.

**DOI:** 10.3390/antibiotics14100973

**Published:** 2025-09-26

**Authors:** Montserrat Muñoz-Rosa, Cristina Elías-López, Rosa Pedraza, Cristina Riazzo, Cristina Arjona-Torres, Isabel Machuca, Rocio Tejero-García, Julian Torre-Cisneros, Luis Martínez-Martínez

**Affiliations:** 1Unit of Microbiology, Reina Sofía University Hospital, 14004 Cordoba, Spain; monserrat.munoz.sspa@juntadeandalucia.es (M.M.-R.); rpedmer@gobiernodecanarias.org (R.P.); cristinal.riazzo.sspa@juntadeandalucia.es (C.R.); rocio.tejero.sspa@juntadeandalucia.es (R.T.-G.); luis.martinez.martinez.sspa@juntadeandalucia.es (L.M.-M.); 2Maimonides Biomedical Research Institute of Cordoba, Reina Sofía University Hospital, University of Cordoba (IMIBIC/HURS/UCO), 14004 Cordoba, Spain; cristina.arjona@imibic.org (C.A.-T.); isabelm.machuca.sspa@juntadeandalucia.es (I.M.); julian.torre.sspa@juntadeandalucia.es (J.T.-C.); 3CIBER de Enfermedades Infecciosas (CIBERINFEC), Instituto de Salud Carlos III, 28029 Madrid, Spain; 4Clinical Unit of Infectious Diseases, Reina Sofía University Hospital, 14004 Cordoba, Spain; 5Department of Agricultural Chemistry, Soil Sciences and Microbiology, University of Cordoba, 14071 Cordoba, Spain

**Keywords:** *Corynebacterium* spp., delafloxacin, antibiotic resistance, *gyrA*

## Abstract

**Background/Objectives:** The susceptibility of *Corynebacterium* spp. to antimicrobial agents is species-related, with increasing levels of resistance to fluoroquinolones in several species related to their continued use in clinical practice. The objectives were to determine the in vitro activity of delafloxacin in comparison with other fluoroquinolones against clinical isolates of *Corynebacterium* spp., to compare MICs of delafloxacin obtained with gradient strips and with reference microdilution, and to investigate the mechanisms related to fluoroquinolone resistance in the tested strains. **Methods:** Fifty-three clinical isolates, assigned to five species of *Corynebacterium* spp., were evaluated using reference microdilution for delafloxacin, ciprofloxacin, levofloxacin, and moxifloxacin (with and without reserpine or phenylalanine-arginine β-naphthylamide), and gradient strips for delafloxacin. The QRDR of the *gyrA* gene was amplified using primers specific to the different species, and mutations were defined after aligning against the corresponding reference sequences. **Results:** Delafloxacin was the most active compound with MIC_50_/MIC_90_ values of 0.5/8 mg/L. Single mutations at the QRDR were observed in isolates, with MICs of delafloxacin ranging from 0.016 to 4 mg/L, while double mutations occurred in isolates, with MICs ranging from 0.125 to 16 mg/L. The delafloxacin gradient strips showed an essential agreement of 88.7%, bias of −5%, and a Kappa index of 0.848. **Conclusions:** Increased MICs of delafloxacin against *Corynebacterium* spp. are related to the presence of non-conservative mutations in the QRDR of *gyrA*. Delafloxacin gradient strips could be a reasonable alternative for use in the clinical routine of the microbiology laboratory. Delafloxacin could represent an alternative for treating infections due to some species of *Corynebacterium*.

## 1. Introduction

*Corynebacterium* species other than *C. diphteriae* are members of the normal microbiota of skin and mucosa, but have also been described as etiologic agents of a variety of human infections [1,2,3]. Susceptibility of *Corynebacterium* spp. to antimicrobial agents is species-related, and several species are frequently resistant to multiple antimicrobial agents [4,5,6]. Glycopeptides and linezolid present excellent in vitro activity against most clinically relevant species [1,4,5,6,7]. β-lactams, fluoroquinolones, aminoglycosides, macrolides, and tetracyclines are also of therapeutic interest for *Corynebacterium* infections [4,5,6].

Previous studies have evaluated the in vitro activity of fluoroquinolones, including ciprofloxacin [6,8,9,10], levofloxacin [6,11], and moxifloxacin [6,10], against *Corynebacterium* spp. As for other organisms, higher rates of resistance to fluoroquinolones in *Corynebacterium* spp. have been observed after the continuous use of fluoroquinolones in clinical practice [8,9,10,11,12]. *Corynebacterium* spp. lack topoisomerase IV [9], and their resistance to fluoroquinolones is primarily linked to point mutations in the quinolone resistance determining region (QRDR) of the *gyrA* gene, encoding the subunit A of DNA gyrase [9,12,13]. The improved activity of delafloxacin against multiple Gram-positive bacteria compared to older quinolones suggests it may also be more active against *Corynebacterium* spp.

Delafloxacin is a new non-zwitterionic dual quinolone, with affinity for both DNA gyrase and topoisomerase IV [14,15]. It presents better in vitro activity against *Staphylococcus aureus* and *Streptococcus pneumoniae* than older quinolones [16]. There is limited information on the activity of delafloxacin against *Corynebacterium* spp. In one study on bacteria cultured from patients with cancer [17], minimal inhibitory concentration (MIC) range, MIC_50_, and MIC_90_ of delafloxacin against 30 *Corynebacterium* spp. were ≤0.03–4, 0.06, and 2 mg/L, much lower than those of ciprofloxacin and levofloxacin. In another recent study [18] on clinical isolates from a single centre (n = 116, 69% *C. striatum* and 16% *C. amycolatum*), MIC range, MIC_50_, and MIC_90_ of delafloxacin were 0.002–>1, 1, and >1 mg/L.

Delafloxacin is not yet included in the antimicrobial panels for automatic susceptibility testing systems commonly used in clinical laboratories, and in vitro testing of this compound is usually performed using discs or gradient strips, although evidence of the reliability of these approaches when considering *Corynebacterium* spp. is lacking.

The objectives of this study were as follows: (i) to determine the in vitro activity of delafloxacin in comparison with ciprofloxacin, levofloxacin, and moxifloxacin against a collection of clinical isolates of *Corynebacterium* spp. using standardized broth microdilution (BMD), (ii) to compare MIC values of delafloxacin obtained with gradient strips with those obtained with BMD, and (iii) to investigate the mechanisms related to fluoroquinolone resistance in the tested strains.

## 2. Results

The MIC range, MIC_50_, and MIC_90_ values obtained with BMD and gradient strips are presented in Table 1. Detailed fluoroquinolone MICs and changes in QRDR of *gyrA* for every isolate are presented in Table 2.

MICs of moxifloxacin were at least two times lower than those of ciprofloxacin for 45/53 (84.9%) isolates, although for *C. glucuronolyticum* MICs of moxifloxacin were the same (4/10 isolates) or two times higher than MICs of ciprofloxacin (3/10 isolates). Similarly, MICs of moxifloxacin were at least four times lower than those of levofloxacin for 42/53 (79.2%). MICs of ciprofloxacin and levofloxacin were within one 2-log dilution step for *C. jeikeium* (10/10 isolates), *C. striatum* (10/10 isolates), and *C. urealyticum* (9/10 isolates), but were lower for ciprofloxacin in the case of *C. glucuronolyticum* (8/10 isolates) and *C. amycolatum* with wild-type *gyrA* (4/4).

Delafloxacin was at least two times more active than the three other tested fluoroquinolones for 50/53 (94.3%) isolates. The remaining three isolates correspond to one *C. jeikeium* with a double mutation in QRDR, and two *C. glucuronolyticum* with wild-type QRDR. For the 53 *Corynebacterium* spp. evaluated (when using reference BMD), 43.4% and 26.4% were inhibited at ≤0.25 mg/L and ≤0.016 mg/L. At ≤0.25 mg/L, the highest percentage of inhibition was observed for *C. amycolatum* (84.6%) and the lowest for *C. striatum* (0%) and *C. urealyticum* (10.0%) (Table 1).

MICs of delafloxacin were just 4, 8, 16, and ≥32 times lower than those of moxifloxacin for 21, 13, 3, and 4 isolates, and for 3 and 1 additional isolates they were ≥4 and ≥8 times lower (Table 2). From another point of view and for the complete set of tested isolates, the MIC_50_ of delafloxacin by the reference method was 8, 16, and 32 times lower than the MIC_50_ of moxifloxacin, ciprofloxacin, and levofloxacin, respectively. Similarly, the MIC_90_ of delafloxacin was 4, >4, and >4 times lower than the MIC_90_ of moxifloxacin, ciprofloxacin, and levofloxacin, respectively (Table 1).

All isolates (except one *C. amycolatum*) for which MICs of delafloxacin were ≤0.016 mg/L had a wild type *gyrA* QRDR. Most *gyrA* QRDR mutations occurred in position 87 (Ser in the five evaluated species) alone or combined with a second mutation in position 91 (Asp in the wild type sequences). In several species, a particular mutation pattern was observed in most isolates (i.e., Ser87Ile in *C. glucuronolyticum*, Ser87Phe plus Asp91Ala in *C. striatum*, or Ser87Val plus Asp91Tyr in *C. urealyticum*). As far as we know, some novel mutations in a concrete species have also been observed, as detailed in Table 2, expanding our information on the importance on this region on resistance of *Corynebacterium* spp. to fluoroquinolones. A point mutation in the QRDR of *gyrA* was related to a moderate increase in MIC values, while double mutations were observed among isolates with MICs of delafloxacin ranging from 0.125 to 16 mg/L (Table 2). This confirms previous observations in *Corynebacterium* spp. [9,12,13].

As shown in Table 2, MICs of delafloxacin for isolates of the same species with just the same *gyrA* QRDR mutation(s) are not the same: i.e., for *C. glucuronolyticum* isolates with the Ser87Ile change, MICs range from 0.25 to 2 mg/L, and for *C. amycolatum* with Ser87Arg, MICs range from 0.016 to 0.125 mg/L, suggesting the presence of additional mechanism(s) of resistance. However, for all tested isolates, MIC values in the presence of the efflux pump inhibitors reserpine or phenylalanine-arginine β-naphthylamide (PAβN) were the same or within one two-fold dilution of those obtained in the absence of the corresponding inhibitor.

The delafloxacin gradient strips showed an essential agreement (EA) of 88.7%, slightly below the threshold of the acceptance criteria (≥90%). Bias was −5% compared to the reference method, which is inside the acceptable range of ±30%. However, the categorical agreement (CA) was 92.5% and the Kappa index was 0.848, indicating an almost perfect concordance. MIC_50_ and MIC_90_ values of delafloxacin by BMD and gradients strips were the same or within one dilution step, except for the MIC_50_ for *C. jeikeium* and the MIC_90_ for *C. urealyticum* (Table 1, Figure 1).

## 3. Discussion

Because of the increasing clinical relevance of multiple species of *Corynebacterium* spp. other than *C. diphtheriae* and their frequent resistance to available antimicrobial agents, it is necessary to evaluate the activity of new compounds against these organisms. Delafloxacin is a new fluoroquinolone of great therapeutic interest because of its dual activity against both DNA gyrase and topoisomerase IV [6,7]. This translates into an improved activity against different Gram-positive bacteria (i.e., *S. aureus*, *Streptococcus pneumoniae*), for which ciprofloxacin and most other fluoroquinolones preferentially target topoisomerase IV [19]. Although *Corynebacterium* spp. lacks topoisomerase IV, delafloxacin is still more active against these microorganisms than the three tested fluoroquinolones. This can be related to the previous observation that, beyond its dual action on DNA gyrase and topoisomerase IV, the main target of delafloxacin for *S. aureus* is, in fact, GyrA, for which delafloxacin is about six times more active than ciprofloxacin [20]. It is likely that targeting DNA gyrase is more efficient than interacting with topoisomerase IV, since the former works ahead of the replication fork to remove DNA supercoils and, consequently, inhibits faster DNA replication. Additionally, delafloxacin differs structurally from other quinolones in presenting a 3-hydroxyazetidine-1-yl substituent at position 7 of the 6-fluoroquinolone core [20], which determines that under acidic conditions the compound is predominantly in a non-ionized form, which improves the activity in acid environments and, in addition, may favor the intrabacterial accumulation of the compound in comparison with other fluoroquinolones.

A large proportion (40/53; 75.5%) of the isolates evaluated in this study were resistant to both ciprofloxacin and moxifloxacin, according to European Committee on Antimicrobial Susceptibility Testing (EUCAST) breakpoints (breakpoints have not been defined for levofloxacin). Delafloxacin is much more active than the tested comparators, in particular against *C. amycolatum*, *C. glucuronolyticum*, and, to a lesser extent, *C. jeikeium*, although MICs of delafloxacin increase in parallel to those of comparative quinolones, as also indicated in a recent report [18]. The worse results were obtained with *C. striatum* and *C. urealyticum*; both species often show high resistance rates to all fluoroquinolones, limiting their use as possible treatment options.

In the absence of clinical breakpoints of delafloxacin for *Corynebacterium* spp. to interpret susceptibility testing data, it is difficult to evaluate the actual impact of its improved in vitro activity from a therapeutic point of view. Considering EUCAST breakpoints for *S. aureus* (susceptible: ≤0.25 mg/L for skin and skin structure infections and ≤0.016 mg/L for community-acquired pneumonia, respectively), only 45.3% and 21.4% of the tested isolates were below these reference breakpoints. As a comparison, 75.5% and 71.7% of the isolates were resistant to ciprofloxacin and moxifloxacin, respectively.

Despite the limited number of studies in which mutations in the QRDR of the *gyrA* gene have been determined in *Corynebacterium* spp., most mutations detected in this study have already been reported. As an example, the Ser87Phe plus Asp91Ala double change frequently observed in *C. striatum* in this study was also the most common pattern in other studies [9,12].

Beyond the clear importance of *gyrA* mutations in compromising the in vitro activity of delafloxacin, the observation that isolates with the same mutations present different MICs of the compound suggests that other mechanisms of resistance modulate delafloxacin resistance in *Corynebacterium* spp. Resistance to fluoroquinolones caused by efflux pumps has been described in *Staphylococcus* spp. and other Gram-positive organisms [21,22], which can be easily demonstrated using active efflux inhibitors [23,24]. The relevance of active efflux in the resistance of *Corynebacterium* spp. to tetracycline and other compounds has also been previously reported [25,26,27]. We hypothesized that efflux pumps could also contribute to fluoroquinolone resistance in our isolates, but we have not observed any relevant difference in the MICs of fluoroquinolones determined in the presence of reserpine or PaβN. This may indicate that active efflux is not really an important mechanism of resistance of the tested isolates against delafloxacin and other fluoroquinolones, or that the inhibitors are not adequate to inhibit such a mechanism, perhaps because their penetration into corynebacteria is limited due to the different surface structure of this microorganism compared to *Staphylococcus* or other Gram-positive bacteria. New studies are warranted to evaluate the importance of active efflux and other mechanisms as a cause of delafloxacin resistance in *Corynebacterium* spp.

Although the reference method for determining the MIC of delafloxacin is broth microdilution, this method is difficult to implement in daily work, and, as this antibiotic is not yet included in most commercial panels of semi-automated microdilution systems, we have explored the reliability of using gradient strips. The obtained results suggest that this approach may represent an acceptable option for testing delafloxacin against *Corynebacterium* spp., although further studies are needed to define categorical errors with the strips when, eventually, specific clinical breakpoints are defined.

The major limitation of this study is that we have focused just on the five species commonly found in our centre, but have not evaluated other species of *Corynebacterium* or related organisms, and that the number of studied isolates was also limited. Further research on these groups of Gram-positive bacteria is also warranted.

## 4. Materials and Methods

### 4.1. Bacterial Strains

Fifty-three clinical isolates (one per patient) of *Corynebacterium* spp., including *C. amycolatum* (n = 13), *C. glucuronolyticum* (n = 10), *C. jeikeium* (n = 10), *C. striatum* (n = 10), and *C. urealyticum* (n = 10), recovered at the clinical microbiology laboratory of the Reina Sofía University Hospital of Cordoba, Spain, and representing the most frequently identified species at our centre during 2018–2022, were evaluated. Species were identified using MALDI-TOF/MS (MALDI Biotyper; Bruker Daltonics, Bremen, Germany).

### 4.2. Susceptibility Testing

MICs of delafloxacin, ciprofloxacin, levofloxacin, and moxifloxacin (all from MedChemExpress, Monmouth Junction, NJ, USA), with and without reserpine (20 mg/L) (MedChemExpress) and PAβN (25 mg/L) (Sigma-Aldrich, Taufkirchen, Germany), were determined by reference BMD (ISO guidelines) [28] using cation-adjusted Mueller–Hinton broth (Sigma-Aldrich) supplemented with 5% lysed horse blood (ThermoFisher, Waltham, MA, USA) and 20 mg/L β-NAD (Sigma-Aldrich) according to the EUCAST recommendations for fastidious organisms. *Streptococcus pneumoniae* ATCC 49619 and *Staphylococcus aureus* ATCC 29213 were used as control strains.

Additionally, susceptibility to delafloxacin was also determined with gradient strips (Etest, BioMèrieux, Rhone, France) using Mueller–Hinton agar with 5% lysed horse blood and 20 mg/L β-NAD (BioMérieux, France). Although EUCAST has not defined clinical breakpoints of delafloxacin for *Corynebacterium*, for analyzing the impact of *gyrA* mutations on the MICs of delafloxacin, the EUCAST breakpoints (V_15.0) [29] for *Staphylococcus aureus* (susceptible: ≤0.25 and ≤0.016 mg/L, for skin and skin structure infections and for community-acquired pneumonia, respectively) were considered.

To evaluate the performance of the gradient strips, EA and bias were calculated according to the recommendations in ISO-20776-2:2021 [30]. Rates of CA were also calculated following the definitions by ISO 20776-2:2007 [31]. The Kappa index was used to evaluate concordance based on Landis and Koch guidelines [32], and was calculated using the Statistical Package for the Social Sciences (IBM SPSS Statistics 19.0, Armonk, NY, USA) software.

### 4.3. Sequencing and Gene Analysis

To amplify the QRDR of the *gyrA* gene of *C. amycolatum*, *C. jeikeium, C. striatum*, and *C. urealyticum*, primers CorynA1: GCGGCTACGTAAAGTCC and CorynA2: CCGCCGGAGCCGTTCAT [9] were used. A novel pair of primers for QRDR of *C. glucuronolyticum* were designed (GluQRDR-F: TACGCATCCTTAATGCCCTG and GluQRDR-R: CCAATCGACCTGAATGAGGA). The amplicons (both strands) were sequenced using Sanger technology (Sistemas Genómicos, Valencia, Spain). Quality control of the sequences was assessed with CLC Genomics Workbench 24.0.3 (Qiagen, Germantown, MD, USA) by manual check and correction of basecalls in the electropherogram and end-trimming. Mutations were defined after BLAST alignment [https://blast.ncbi.nlm.nih.gov/Blast.cgi (accessed on 1 January 2025)] against the following corresponding reference sequences: *C. amycolatum* quinolone-susceptible clinical isolate [9] (Accession number NCBI: AY559039), *C. glucuronolyticum* strain FDAARGOS_1053 (Accession number NCBI: NZ_CP066007.1), *C. jeikeium* ATCC 43734 (Accession number NCBI: NZ_GG700813.1), *C. striatum* ATCC 6940 (Accession number NCBI: AY559038), and *C. urealyticum* DSM 7109 [33] (Accession number NCBI: NC_010545.1).

## 5. Conclusions

In summary, compared to the other fluoroquinolones analyzed, delafloxacin was the most in vitro active compound against *Corynebacterium* spp. It may represent a therapeutic alternative for infections caused by these organisms, and could help preserve the efficacy of fluoroquinolones through targeted therapy. The clinical implications of these findings warrant further study, including clinical efficacy and risk of emergence of new resistance.

The increase in the MICs of fluoroquinolones against *Corynebacterium* spp. is related to non-conservative mutations in the QRDR of the *gyrA* gene of the studied isolates. The level of increase depends on whether there are one or two mutations, their positions, and the considered species. Indirect evidence suggests that additional mechanisms contribute to decreased activity of delafloxacin against *Corynebacterium* spp.

Gradient strips may represent a reasonable strategy for determining the MIC of delafloxacin against *Corynebacterium* spp. in routine work.

## Figures and Tables

**Figure 1 antibiotics-14-00973-f001:**
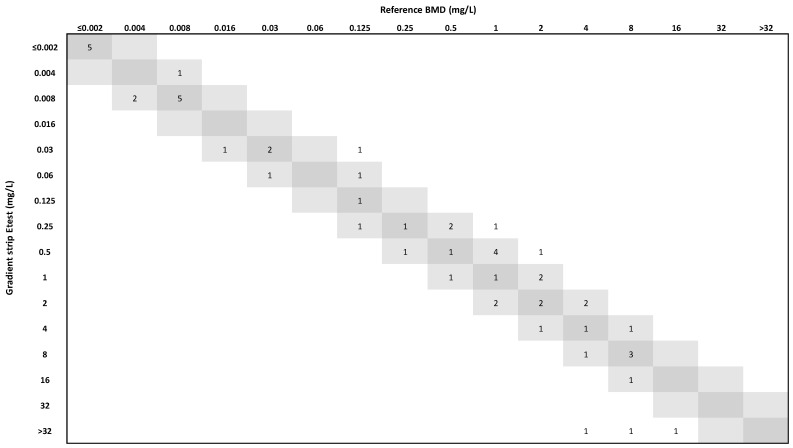
Correlation between delafloxacin MICs as determined by reference broth microdilution (BMD) and Etest. Strains with MICs corresponding to BMD and 1-log_2_ dilution are indicated in the dark and light grey squares, respectively.

**Table 1 antibiotics-14-00973-t001:** In vitro activity of delafloxacin determined by reference broth microdilution (BMD) or gradient strips (GS), and of three other fluoroquinolones by BMD against clinical isolates of *Corynebacterium* spp.

			DELAFLOXACIN					CIPROFLOXACIN				LEVOFLOXACIN			MOXIFLOXACIN			
Species	N		MIC ^a^ ≤0.25	MIC ≤0.016	MIC Range	MIC_50_	MIC_90_	MIC Range	MIC_50_	MIC_90_	R ^b^ (N, %)	MIC Range	MIC_50_	MIC_90_	MIC Range	MIC_50_	MIC_90_	R ^b^ (N, %)
*C.* *amycolatum*	13	BMD	11 (84.6%)	5 (38.5%)	≤0.002–2	0.03	0.5	0.016–>32	4	>32	9 (69.2%)	0.06–>32	4	32	0.016–16	1	8	8 (61.5%)
		GS	12 (92.3%)	4 (30.8%)	≤0.002–2	0.023	0.25	-	-	-	-	-	-	-	-	-	-	-
*C.* *glucuronolyticum*	10	BMD	4 (40%)	3 (30%)	≤0.002–2	0.5	2	0.016–4	4	4	7 (70%)	0.06–>32	16	32	0.004–8	4	8	7 (70%)
		GS	4 (40%)	3 (30%)	≤0.002–1	0.38	1	-	-	-	-	-	-	-	-	-	-	-
*C. jeikeium*	10	BMD	7 (70%)	5 (50%)	0.004–16	0.008	8	0.06–>32	0.125	>32	5 (50%)	0.125–>32	0.25	>32	0.03–16	0.06	16	4 (40%)
		GS	6 (60%)	4 (40%)	0.006–>32	0.19	8	-	-	-	-	-	-	-	-	-	-	-
*C. striatum*	10	BMD	0(0%)	0 (0%)	0.5–4	1	4	4–>32	32	>32	10 (100%)	4–>32	>32	>32	2–8	8	8	10 (100%)
		GS	1 (10%)	0 (0%)	0.25–3	1.5	3	-	-	-	-	-	-	-	-	-	-	-
*C. urealyticum*	10	BMD	1 (10%)	1 (10%)	0.008–8	4	8	0.06–>32	>32	>32	9 (90%)	0.25–>32	>32	>32	0.06–32	32	32	9 (90%)
		GS	1 (10%)	1 (10%)	0.006–>32	6	>32	-	-	-	-	-	-	-	-	-	-	-
*Corynebacterium* spp.	53	BMD	23 (43.4%)	14 (26.4%)	≤0.002–16	0.5	8	0.016–>32	8	>32	40 (75.5%)	0.06–>32	16	>32	0.004–32	4	32	38 (71.7%)
		GS	24 (45.3%)	12 (21.4%)	≤0.002–>32	0.5	6	-	-	-	-	-	-	-	-	-	-	-

^a^ MIC values expressed in mg/L. ^b^ R: Resistant, according to EUCAST breakpoints (ciprofloxacin > 1 mg/L, moxifloxacin > 0.5 mg/L).

**Table 2 antibiotics-14-00973-t002:** Relationship between MICs (mg/L) of delafloxacin (DEL), moxifloxacin (MOX), ciprofloxacin (CIP), and levofloxacin (LEV), and mutations in the QRDR of *gyrA* of five *Corynebacterium* species.

ID_Isolates	DEL	MOX	CIP	LEV	Amino Acid Change(s) ^a^
*C. amycolatum*	NA ^b^	NA	NA	NA	Ser-Ala-Asp (87–88–91)
CHURS-201026	≤0.002	0.016	0.016	0.06	WT
CHURS-201094	≤0.002	0.008	0.016	0.06	WT
CHURS-202160	≤0.002	0.008	0.016	0.06	WT
CHURS-201550	≤0.002	0.008	0.016	0.03	WT
CHURS-200784	0.016	1	2	4	Ser87Arg
CHURS-200921	0.03	1	4	4	Ser87Arg
CHURS-201216	0.03	1	4	4	Ser87Arg
CHURS-202276	0.03	1	4	4	Ser87Arg
CHURS-201445	0.125	0.5	8	4	Ser87Arg
CHURS-202275	0.125	1	8	8	Ser87Ile
CHURS-201033	0.125	2	32	16	Ser87Ile Ala88Val *
CHURS-201637	0.5	8	>32	32	Ser87Arg Ala88Pro
CHURS-202078	2	16	>32	>32	Ser87Ile Asp91Gly
*C. glucuronolyticum*	NA	NA	NA	NA	Ser (87)
CHURS-201459	≤0.002	0.004	0.016	0.06	WT
CHURS-200111	0.008	0.06	0.125	0.125	WT
CHURS-201458	0.008	0.06	0.125	0.25	WT
CHURS-202161	0.25	4	4	16	Ser87Ile
CHURS-201434	0.5	4	4	32	Ser87Ile
CHURS-200113	0.5	4	4	16	Ser87Ile
CHURS-200326	1	8	4	32	Ser87Ile
CHURS-200692	1	8	8	>32	Ser87Ile
CHURS-200112	2	8	4	32	Ser87Ile
CHURS-201388	2	8	4	32	Ser87Ile
*C. jeikeium*	NA	NA	NA	NA	Ser-Asp (87–91)
CHURS-182337	0.004	0.03	0.06	0.125	WT
CHURS-222505	0.004	0.03	0.06	0.125	WT
CHURS-200421	0.008	0.03	0.06	0.125	WT
CHURS-201636	0.008	0.03	0.06	0.25	WT
CHURS-181588	0.008	0.06	0.125	0.25	WT
CHURS-220538	0.125	1	4	4	Ser87Arg
CHURS-205744	0.25	2	16	8	Ser87Arg
CHURS-181717	1	2	8	16	Ser87Ile
CHURS-183397	8	16	>32	>32	Ser87Ile Asp91Tyr
CHURS-192312	16	16	>32	>32	Ser87Ile Asp91Tyr
*C. striatum*	NA	NA	NA	NA	Ser-Asp (87–91)
CHURS-201982	0.5	2	4	4	Ser87Val
CHURS-200172	1	4	16	32	Ser87Phe Asp91Ala
CHURS-200532	1	4	16	32	Ser87Phe Asp91Ala
CHURS-200609	1	4	32	>32	Ser87Phe Asp91Ala
CHURS-205232	1	8	>32	>32	Ser87Phe Asp91Ala
CHURS-201900	2	8	32	>32	Ser87Phe Asp91Ala
CHURS-201917	2	8	32	>32	Ser87Phe Asp91Ala
CHURS-202400	2	8	>32	>32	Ser87Phe Asp91Ala
CHURS-202043	4	8	32	>32	Ser87Phe Asp91Ala
CHURS-202253	4	8	>32	>32	Ser87Phe Asp91Ala
*C. urealyticum*	NA	NA	NA	NA	Ser-Asp (87–91)
CHURS-200585	0.008	0.06	0.06	0.25	WT
CHURS-201308	1	4	32	16	Ser87Val
CHURS-200529	4	16	>32	>32	Ser87Val Asp91Tyr
CHURS-201820	4	32	>32	32	Ser87Val Asp91Tyr
CHURS-200825	4	32	>32	>32	Ser87Val Asp91Tyr
CHURS-200751	8	32	>32	>32	Ser87Val Asp91Tyr
CHURS-201390	8	32	>32	>32	Ser87Val Asp91Tyr
CHURS-201559	8	32	>32	>32	Ser87Val Asp91Tyr
CHURS-201661	8	32	>32	>32	Ser87Val Asp91Tyr
CHURS-201808	8	32	>32	>32	Ser87Tyr Asp91Phe *

^a^ Amino acids in the positions of the wild-type (WT) sequence of the reference strains are also shown. ^b^ NA: not available. * Novel mutation.

## Data Availability

The original contributions presented in this study are included in the article. Further inquiries can be directed to the corresponding author(s).

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
