# Peer review of "In Vitro Activity of Delafloxacin Against *Corynebacterium* spp."

_antibiotics, 2025, doi:10.3390/antibiotics14100973_

Round 1
Reviewer 1 Report
Comments and Suggestions for Authors
Comment 1. The abstract contains excessive methodological and numerical details. I suggest authors to condense MIC ranges and focusing instead on the novelty and implications of the findings.
Comment 2. The novelty of the work is not explicitly stated. Please clearly state why delafloxacin’s activity against Corynebacterium spp. is important in the current antimicrobial resistance landscape.
Comment 3. The conclusion of abstract section is more descriptive rather than interpretive. I would suggest the authors to refine the last lines to highlight clinical relevance and potential changes to current practices.
Comment 4. In the Introduction section, the problem statement is diluted within the literature review. Please write the core rationale for the study into a clear, concise statement early in the introduction section.
Comment 5. In methodology section, the sample size of 53 isolates has not been justified with a clear rationale or statistical reasoning for the number of isolates.
Comment 6. The details of sequencing are lacking the mention of quality controls.
Comment 7. Data presentation is highly quite good but lacks prioritization. I would suggest the authors to restructure the results and lead with the most clinically relevant findings before detailing species-specific MIC ranges.
Comment 8. No statistical significance testing is applied to MIC differences between species or mutation categories. Please revise the results after performing the basic statistical comparisons to strengthen conclusions.
Comment 9. Mutation data has been comprehensively presented but it would be better to correlate these mutations with resistance magnitude for clinical interpretability.
Comment 10. The discussion could be improved by adding a more critical risk–benefit framing.
Comment 11. In conclusion section, the impact of this study on antimicrobial stewardship is not addressed. Please state how these results could help preserve fluoroquinolone efficacy through targeted use.
Reviewer 2 Report
Comments and Suggestions for Authors
The study thoroughly assesses delafloxacin's activity against five clinically relevant Corynebacterium spp., comparing it with other fluoroquinolones and investigating resistance mechanisms. The study expands knowledge on delafloxacin’s efficacy and resistance patterns in Corynebacterium spp., a less-studied area. Some minor justification are required
- The introduction lists objectives but lacks a clear hypothesis or rationale for why delafloxacin might outperform other fluoroquinolones in Corynebacterium spp. ?
- The agreement between BMD and GS is reported (EA: 88.7%, CA: 92.5%), but no statistical tests (e.g., Bland-Altman analysis, kappa coefficient) are provided to validate these results.
- The discussion briefly mentions the lack of clinical breakpoints for delafloxacin but does not explore other limitations (e.g., small sample size for some species, single-center isolates).
- The conclusion is brief and does not outline specific next steps like efficacy in animal models or clinical cases, exploring combination therapies to overcome resistance. etc.,
By addressing the above points particularly expanding on resistance mechanisms, improving statistical rigor, and clarifying clinical relevance the manuscript could significantly enhance its scientific impact and utility for clinicians and researchers.
Reviewer 3 Report
Comments and Suggestions for Authors
The manuscript "In vitro activity of delafloxacin against Corynebacterium spp." addresses an important and timely topic: delafloxacin's comparative in vitro activity against clinically relevant Corynebacterium species. Given the increasing clinical relevance of non-diphtheriae Corynebacterium spp. and the limited treatment options available, the study targets an underexplored area. The focus on delafloxacin as a potential alternative is timely and of clinical interest. The detailed mapping of QRDR mutations in gyrA and their correlation with MIC variations enhances the study's scientific value. The work is well-structured, methodologically sound, and supported by precise experimental data. The authors comprehensively analyze antimicrobial susceptibility patterns, incorporating reference broth microdilution and gradient strip methods and genetic characterization of gyrA mutations associated with fluoroquinolone resistance. Moreover, identifying previously reported and novel mutations supports the authors’ hypotheses and expands the knowledge base on Corynebacterium spp resistance mechanisms. This added genotypic layer of evidence significantly enhances the manuscript’s scientific value and impact, making the findings more robust and potentially helpful in guiding future research and surveillance.
Overall, this is a well-conducted and informative study that provides valuable insights into the potential role of delafloxacin in treating infections caused by Corynebacterium spp. The manuscript could be further strengthened by expanding the clinical context of the findings, providing more detailed species-specific comparisons, and refining data presentation. After addressing these points, the work will contribute to the AMR and therapy field.
- Please italicize all gene names (Please check the Results section).
- Please italicize all bacterial names (Please check the Results section).
- Please write out numbers less than ten in words.
- While the in vitro activity of delafloxacin is demonstrated, there is an absence of established EUCAST breakpoints for Corynebacterium, limiting direct therapeutic translation. The discussion could be strengthened by elaborating on possible clinical implications and potential breakpoints extrapolated from other Gram-positive organisms.
- Although species present MIC data, comparing resistance patterns and potential species-related intrinsic factors influencing delafloxacin activity more explicitly would enhance interpretability.
Round 2
Reviewer 1 Report
Comments and Suggestions for Authors
Authors have addressed all the suggestions and have significantly improved the quality of manuscript. Now I recommend the manuscript to publish in its present form.